# Change in Alcohol Use during the Prolonged COVID-19 Pandemic and Its Psychosocial Factors: A One-Year Longitudinal Study in Japan

**DOI:** 10.3390/ijerph20053871

**Published:** 2023-02-22

**Authors:** Nagisa Sugaya, Tetsuya Yamamoto, Naho Suzuki, Chigusa Uchiumi

**Affiliations:** 1Department of Public Health, School of Medicine, Yokohama City University, Yokohama 236-0004, Japan; 2Graduate School of Technology, Industrial and Social Sciences, Tokushima University, Tokushima 770-8502, Japan; 3Graduate School of Sciences and Technology for Innovation, Tokushima University, Tokushima 770-8502, Japan

**Keywords:** alcohol use, COVID-19, longitudinal study

## Abstract

This study investigated changes in alcohol use and its related psychosocial factors during the COVID-19 pandemic in Japan. Two online surveys were completed by participants between 15 and 20 June 2021 (phase 1) and 13 and 30 May 2022 (phase 2). A total of 9614 individuals participated in both phases (46% women, mean age = 50.0 ± 13.1 years) and a repeated three-way analysis of variance and multinomial logistic regression analysis were conducted. These data analyses showed that the presence of hazardous alcohol use at phase 2 was predicted by being male and unmarried, having a higher annual household income and age, having a larger social network, and displaying fewer COVID-19 prevention behaviors at phase 1. Further, the presence of potential alcoholism at phase 2 was predicted by being male, being more anxious, having a larger social network, exercising more, showing a deterioration of economic status, having more difficulties owing to a lack of daily necessities, having less healthy eating habits, and showing fewer COVID-19 prevention behaviors at phase 1. These findings suggest that psychological problems and increased work (or academic) and economic difficulties were associated with severe alcohol problems during a later stage of the COVID-19 pandemic.

## 1. Introduction

After its outbreak in December 2019, the coronavirus disease 2019 (COVID-19) spread rapidly worldwide [1]. To deter its spread, many countries imposed repeated lockdowns such as restricting people’s movement and temporarily closing services. However, while lockdowns are effective at preventing the spread of infection, they cause not only significant psychological distress [2,3], but also financial hardship.

Many studies conducted in various countries have examined the consumption of alcohol during the COVID-19 pandemic and its lockdown periods [4,5]. During the pandemic, alcohol use was reported to be a factor that increased the probability of suicidal behaviors [6]. Some studies in India, the United States, and the Czech Republic found a decrease [7] or no change [8,9] in alcohol-related problems during the COVID-19 pandemic; however, other research in the United States and Norway showed an increase in alcohol-related issues [10,11,12]. Moreover, the findings from eight European countries indicated that during the initial months of the COVID-19 pandemic, the top 10% of drinkers’ alcohol consumption and prevalence of heavy drinking episodes increased, whereas alcohol consumption in the sample’s other groups decreased [13]. Considering that alcohol use is a major risk factor for communicable, maternal, perinatal, nutritional, and non-communicable diseases as well as injuries and deaths, with an estimated three million alcohol-induced deaths worldwide in 2016 [14], it is important to better understand alcohol use to protect people’s health.

Between April 2020 and August 2021, the Japanese government declared four states of emergency owing to the COVID-19 pandemic. However, while many countries imposed lockdowns with penalties for violations, Japan’s COVID-19 policy was different in that it merely requested people to refrain from leaving home, except in the case of emergencies, and placed temporary restrictions on certain businesses—without inflicting penalties. However, this “mild lockdown” [15] in Japan still affected people’s lives in many ways, including lifestyle changes due to remote working and online learning as well as financial hardship due to reduced income and unemployment [16]. Indeed, the prolonged COVID-19 pandemic and repeated declarations of a state of emergency were responsible for maintaining the unfavorable psychological state of the Japanese population [17,18].

These long-term stressful conditions and behavioral restrictions might have affected the Japanese population’s alcohol use. In a recent survey in Japan, although the percentage of men who drank alcohol in amounts that increased their risk of lifestyle-related diseases (women: 20 g/day or more of pure alcohol; men: 40 g/day or more) decreased between 2013 and 2018, it remained unchanged for women [19]. Further, a 2018 nationwide survey in Japan suggested that alcoholism was prevalent among 40,000 women and 220,000 men, of whom around 80% did not seek medical care [19].

In a survey conducted in June 2021 [20], during the midst of the COVID-19 pandemic, the prevalence of potential alcoholism among the study participants was higher than previously reported, particularly among women. The presence of potential alcoholism was associated with the worsening of peoples’ psychological state, especially depression and anxiety, and various difficulties in daily life due to the COVID-19 pandemic. By contrast, a previous investigation [21] on the dietary habits among the Japanese population conducted during the COVID-19 pandemic’s early phase (January to May 2020) reported a decrease in alcohol intake. These mixed findings could be because of the different samples used in these studies or discrepancies in the population’s drinking status depending on the times at which they were conducted.

Although a state of emergency has not been declared in Japan since October 2021, the spread of COVID-19 has continued (Figure 1). Therefore, it is important to identify which individuals are most likely to develop risky drinking behaviors during a pandemic and what interventions they should be offered to provide useful information for establishing effective approaches to mental health problems in such a situation. Because research evaluating the longitudinal changes in alcohol use problems during a pandemic in Japan is insufficient, this study bridges an important gap in the literature.

This study investigated changes in drinking status and the related psychosocial factors approximately one year after the June 2021 survey [20]. The large sample size included in this study and one-year follow-up design will help determine the status of alcohol-related problems during the prolonged COVID-19 pandemic in Japan, which has not been clarified in previous studies. Further, our study is expected to identify the characteristics related to alcohol use to help the early prevention and follow-up of alcohol-related problems.

## 2. Materials and Methods

### 2.1. Participants and Data Collection

The online surveys were conducted between 15 and 20 June 2021 (phase 1) and between 13 and 30 May 2022 (phase 2). The inclusion criteria were as follows: (a) inhabitants living in six prefectures (Tokyo, Aichi, Osaka, Kyoto, Hyogo, and Fukuoka) and (b) age ≥ 20 years at phase 1. The exclusion criterion was age < 20 years. This study followed up on those participants who were residents in the target areas when the first (May 2020) and second (February 2021) states of emergency were declared. The survey period of phase 1 was during the third state of emergency declaration in the areas covered by this study. Phase 1 was approximately one year after the COVID-19 pandemic effects became significant in Japan (the first state of emergency declaration). Since the Alcohol Use Disorders Identification Test (AUDIT) used to examine the drinking problems in this study enquires about alcohol consumption in the past year, phase 1 was thus the best time to assess alcohol use during the pandemic.

This study’s participants were recruited by Macromill, Inc. (Tokyo, Japan); this global marketing research company has over 1.3 million registered members from all prefectures in Japan, with diverse characteristics regarding sex and age. Registered members who lived in the target prefectures were recruited through email; additionally, their data were collected using an online platform. The participants completed the online survey after having received the link. All the participants voluntarily responded to the survey anonymously and provided their informed consent online before completing it. They were clearly informed about the survey procedure and understood that they could interrupt or terminate the survey at any time without having to provide a reason. If any item was left unanswered, the questionnaire format, excluding the default items provided by Macromill, Inc. (sex, age, occupation, annual household income, marital status, and presence of children), did not allow the participants to proceed to the subsequent page. All the participants were rewarded with Macromill points, a points-based reward system provided by Macromill. Inc., which could be exchanged for prizes or cash.

This study was approved by the Research Ethics Committee of the Graduate School of Social and Industrial Science and Technology, Tokushima University (acceptance number 212); further, it was performed according to the ethical standards of the 1964 Declaration of Helsinki and its later amendments.

### 2.2. Measurements

#### 2.2.1. Sociodemographic Characteristics

We collected the participants’ sociodemographic information including age, sex, employment status (employed, homemaker, student, unemployed, or other), marital status, the presence of children, and annual household income (<2.0, 2.0–3.9, 4.0–5.9, 6.0–7.9, or ≥8.0 million yen, or unknown).

#### 2.2.2. Alcohol Use

Alcohol use, the main outcome of this study, was assessed using the Japanese version of the AUDIT [22]. It consists of 10 items across three domains: hazardous alcohol use, dependence symptoms, and harmful alcohol use (three, three, and four items, respectively). Each item was scored on a scale from zero to four. The lowest and highest possible scores of the AUDIT were 0 and 40, respectively. A higher score indicated an increased likelihood and severity of hazardous drinking, harmful drinking, and alcohol dependence. Those individuals who scored 8–14 and ≥15 points were considered to have a hazardous drinking problem (hazardous alcohol use group) and show alcohol dependence (potential alcoholism group), respectively, based on the AUDIT’s cut-off criteria provided by the World Health Organization and health guidance by the Japanese Ministry of Health, Labour and Welfare [23,24]. The participants who scored ≤ 7 points were defined as those without alcohol problems (no alcohol problem group). Previous research conducted in Japan before the COVID-19 pandemic reported that men had higher AUDIT scores than women [25].

#### 2.2.3. Psychological Distress

We used the Japanese version of the Kessler Psychological Distress Scale-6 (K6) [26], which is a six-item screening tool measuring non-specific psychological distress over the past 30 days. Each question was rated on a five-point Likert scale from zero (never) to four (always), with the total score ranging from 0 to 24. Because of its high accuracy and conciseness, the K6 is considered as an ideal instrument for screening mental disorders or psychological distress in population-based health surveys [26,27,28]. K6 scores ranging from 5 to 12 were defined as mild-to-moderate psychological distress. This is the optimal lower threshold for assessing moderate psychological distress [29]. Previous studies have traditionally employed a threshold score of 13 points [27,30]. K6 scores of 13 points or higher and of 4 points or lower were defined as serious and no/low psychological distress, respectively.

#### 2.2.4. Depression Symptoms

We used the Japanese version of the Patient Health Questionnaire-9 (PHQ-9) [31] to assess the symptoms of depression; it comprises nine questions. The participants reported their symptoms of depression during the past two weeks on a scale from zero (not at all) to three (nearly every day) [32]. A score of 10 points or higher indicated a high likelihood of major depression.

#### 2.2.5. Anxiety

We used the Japanese version of the Generalized Anxiety Disorder-7 (GAD-7) [33] to assess anxiety symptoms during the past two weeks. Seven questions were rated on a scale from zero (never) to three (almost every day). The total score ranged from 0 to 21 points [34], with 0 to 4, 5 to 9, 10 to 14, and 15 to 21 points indicating minimal, mild, moderate, and severe anxiety, respectively. A respondent who scored 10 points or more was considered as requiring drug therapy.

#### 2.2.6. Loneliness

We employed the Japanese version of the University of California, Los Angeles, Loneliness Scale Version 3 (UCLA-LS3) [35] to assess loneliness during the state of emergency (phase 1) and during the past 30 days (phase 2). It consists of 10 items, each rated on a scale from one (never) to four (always) [36]. The total score ranged from 10 to 40, with higher scores indicating greater levels of loneliness.

#### 2.2.7. Social Isolation

The Japanese version of the abbreviated Lubben Social Network Scale (LSNS-6) [37] was applied to measure the social networks of the participants during phases 1 and 2. It includes six items on the networks of friends and family who provide emotional and vital support; three items are related to friendship networks and three items to family networks. All the questions were rated on a scale from zero (none) to five (nine or more) [38]. The total score ranged from 0 to 30 points, with higher scores indicating a larger social network; furthermore, scores below 12 points indicated social isolation.

#### 2.2.8. COVID-19-Related Lifestyle Changes, Coping Behavior, and Stressors

With extensive reference to the COVID-19 literature [39,40,41,42,43], we developed items related to lifestyle changes and coping behaviors (eight items) and stressors (seven items) during the pandemic. We requested the participants to rate the frequency and experience of these items during phases 1 and 2 on a scale from one (not at all) to seven (extremely). Since these items have not been validated, they were scored separately rather than relying on the total score.

### 2.3. Statistical Analysis

As noted above, the AUDIT scores of phases 1 and 2 were classified into three groups based on cut-off points: the no alcohol problem, hazardous alcohol use, and potential alcoholism groups. A paired *t* test was applied to compare the AUDIT scores, psychological indexes, and COVID-19 pandemic-related variables between phases 1 and 2. Moreover, a chi-squared test comparing the sociodemographic data between the three AUDIT groups was performed in each phase. Additionally, a repeated three-way analysis of variance (ANOVA) was conducted to confirm the interactions among the three AUDIT groups at phases 1 and 2 and among them when considering the psychological indexes and COVID-19 pandemic-related variables. Multinomial logistic regression analyses using the backward selection method were employed to examine the effects of the sociodemographic characteristics, psychological indexes, and COVID-19 pandemic-related variables at phase 1 on hazardous alcohol use and potential alcoholism at phase 2. (The three AUDIT groups at phase 1 were treated as adjustment variables.) Furthermore, the multicollinearity among the independent variables included in the final model was checked to assess potential bias in the results due to collinearity. For all the two-tailed tests, the significance was set at *α* = 0.05. The statistical analyses were performed using SPSS (version 25.0; IBM, Armonk, NY, USA).

## 3. Results

### 3.1. Descriptive Statistics

Table 1 shows the sociodemographic characteristics of our sample. At phase 2, 11,427 individuals had participated, and a follow-up survey was conducted. Overall, 9614 individuals participated in phases 1 and 2 (46% women, mean age = 50.0 ± 13.1 years, range = 20–90 years); thus, 1813 (15.9%) people who participated at phase 1 did not respond in phase 2.

In our sample, there were no missing data for any of the variables except annual household income, for which 908 participants (9.4%) did not provide their income. Therefore, the “unknown” classification for annual household income included missing values (*N* = 1037). Table 2 shows the number of the participants assigned to the three AUDIT groups in each phase.

### 3.2. Differences in the AUDIT Scores, Psychological Indexes, and COVID-19 Pandemic-Related Variables between Each Phase

Table 3 shows the differences in each variable between the phases. The PHQ-9 and GAD-7 scores at phase 2 were significantly lower than those at phase 1 (*p* < 0.05); however, the effect sizes did not exceed the lower limit for a small effect size (i.e., Cohen’s d > 0.200). All the COVID-19 pandemic-related variables except “healthy sleep habits” and “Continuous prevention behaviors of COVID-19,” showed significant differences between the phases (*p* < 0.05); nevertheless, only the effect size of “Offline interaction with familiar people” exceeded the lower limit for a small effect size.

### 3.3. Differences in the Sociodemographic Characteristics between the Three AUDIT Groups in Each Phase

Table 1 shows the differences in the sociodemographic characteristics between the three AUDIT groups in each phase. There were significant differences between the three AUDIT groups for all the sociodemographic characteristics (*p* < 0.05); however, only the effect size for sex exceeded the lower limit for a small effect size (Cramer’s *V* > 0.100).

### 3.4. Differences and Interactions between the Phases and AUDIT Groups for the Psychological Indexes and COVID-19 Pandemic-Related Variables

Table 4 and Table 5 display the differences and interactions between the phases (1 and 2) and AUDIT groups (no alcohol problem, hazardous alcohol use, and potential alcoholism) for the psychological indexes and COVID-19 pandemic-related variables. Table 4 shows the means and standard deviations of those variables by group for each phase. Regarding the interactions between the phases and groups (Table 5), the results for the K6 items of “Offline interaction with familiar people” and “Difficulties in work or schoolwork” were significant. All the other variables except “Exercise” showed only significant main effects for the phase or group. Simple main effect tests were conducted for those variables that showed significant interactions (Figure 2). The AUDIT group at phase 1 had significant effects on all the variables except for “Exercise” and “Offline interaction with familiar people”. At phase 2, it had significant effects on all the psychological indexes except the LSNS-6 and all the COVID-19 pandemic-related variables except “Exercise”, “Favorite activity”, “Offline interaction with familiar people”, and “Optimism”.

### 3.5. Psychosocial Factors Relating to Hazardous Alcohol Use and Potential Alcoholism a Year Later

Table 6 shows the results of the final multinomial logistic regression analysis examining the psychosocial factors at phase 1 relating to the presence of hazardous alcohol use and potential alcoholism at phase 2. No multicollinearity problems were found among the independent variables (all variance inflation factors were under 4.46). The variables at phase 1 that were significantly related to the presence of hazardous alcohol use at phase 2 included the male gender, unmarried status, higher annual household income, higher age, greater LSNS-6 score, and lower continuous prevention score. The variables at phase 1 that were significantly related to the presence of potential alcoholism at phase 2 comprised the male gender, higher GAD-7 and LSNS-6 scores, higher exercise scores, a deterioration of economic status, more difficulties owing to a lack of daily necessities, and lower scores for healthy eating habits and continuous prevention.

## 4. Discussion

The distribution of the number of participants in the potential alcoholism, hazardous alcohol use, and no alcohol problem groups did not differ significantly between the two phases. About half of the individuals with hazardous alcohol use and those with potential alcoholism at phase 2 were assigned to other AUDIT groups at phase 1. Hence, as the level of alcohol consumption problems varied during the pandemic, exploring relevant factors may help identify the causes of the deterioration of mental health and propose intervention strategies. We examine the relevant factors in the following subsections.

### 4.1. Comparisons by Sociodemographic Characteristics

Overall, there was a similar trend to the pre-pandemic data in that more men presented hazardous alcohol use and potential alcoholism than women. However, compared with the pre-pandemic data [19], the proportions of these groups were higher, especially among women. Regarding age, the prevalence of hazardous alcohol use and potential alcoholism was higher in the 50–64 age group than in the other groups at both phases. A Japanese epidemiological study before the pandemic [44] reported the highest proportion of hazardous alcohol use among men in their 50s; moreover, in this research’s data, the high prevalence of hazardous alcohol use among men in the 50–64 age group was more pronounced than that for women, similar to in previous studies.

In the comparisons among occupations, the prevalence of hazardous alcohol use and potential alcoholism in the employee group was higher than that in the other occupation groups. Although there was no remarkable change in the prevalence between the phases, it was higher than that reported before the COVID-19 pandemic. A previous study in Japan [45] found that during a pandemic, the fewer opportunities to leave home and more time spent at home are factors that contribute to an increase and decrease in alcohol consumption, respectively. Other research in Japan has reported that unwanted remote working during a pandemic is associated with increased alcohol consumption [46]. However, a detailed analysis of the factors influencing the development and maintenance of problematic drinking behaviors among employees during a pandemic is needed.

Regarding marital status and the presence of children, hazardous alcohol use was more common among married people and those with children at both phases. At phase 2, potential alcoholism was also more common among married individuals. However, the results did not differ substantially from those at phase 1; thus, they should be interpreted with caution. Finally, regarding household income, hazardous alcohol use and potential alcoholism were more prevalent among those with higher household incomes at both phases. Although issues such as unemployment due to the pandemic have been discussed, these results indicate that financial difficulties and poverty are not necessarily the only factors directly related to problematic drinking.

These results suggest that we may be unable to formulate strategies for alcohol-related problems during a pandemic based solely on the results of studies conducted before the COVID-19 outbreak, not to mention our stereotypes. The exacerbation of alcohol-related problems may be more pronounced among women than men, while employed and higher-income individuals, who are considered to be relatively financially stable, as well as individuals living with their families, who are seemingly less isolated, should be alerted to the possible exacerbation of alcohol-related problems during a pandemic.

### 4.2. Comparisons of the COVID-19 Pandemic-Related Variables and Psychological Indexes between the Phases

Across all the participants, there were significant increases in depression and anxiety, lifestyle changes, and many of the COVID-19 pandemic-related variables from phase 1 to phase 2. This included interactions with others, optimism, economic status, relationships, frustration, anxiety, sleep problems, insufficient daily necessities, and difficulties with work and schoolwork; however, several of the psychological indexes fell compared with pre-pandemic studies. This finding suggests that even two years after the start of the COVID-19 pandemic, mental health deterioration remains a serious problem. This may be related to alcohol-related problems in groups with seemingly no socioeconomic risk factors, as discussed in the previous section.

### 4.3. Interactions between the AUDIT Groups and Phases

A detailed analysis of the interactions between the AUDIT groups and phases showed differences in the participants’ characteristics that were not apparent in the baseline analysis. First, there was a significant interaction between the AUDIT groups and phases for the K6 score. Regarding the individuals with no alcohol problem at phase 1, those who developed potential alcoholism at phase 2 showed more severe psychological distress at each phase than the other groups as well as a significant increase in psychological distress between the phases. Hence, even those who did not have an alcohol problem at phase 1 were mentally vulnerable and developed serious alcohol-related problems for some reason. Among the individuals with hazardous alcohol use at phase 1, those who recovered and had no alcohol problem at phase 2 showed significantly lower psychological distress; moreover, only those who maintained hazardous alcohol use throughout the two phases had lower psychological distress than those with worsened or improved alcohol use at phase 2. The people who developed hazardous drinking behavior to the point of addiction already had high psychological distress at phase 1, while those who maintained it had the lowest psychological distress at any point in time. The latter may represent drinking for pleasure, unrelated to stress, based on the transtheoretical model of health behavior change; hence, such individuals may be in the periods of pre-contemplation, contemplation, or preparation, which occurs when they are aware of the risks between phases 1 and 2 but have not yet reduced their drinking. An intervention may also be necessary for those individuals who do not demonstrate such a marked psychological change.

Regarding the individuals showing potential alcoholism at phase 1, those who recovered and had no alcohol problem showed the highest psychological distress at phase 1 compared with the other groups; additionally, they showed a prominent increase in psychological distress, while those who maintained potential alcoholism showed lower such distress. Furthermore, those who recovered and had no alcohol problem could have been exposed to a temporary high level of stress that caused their drinking to reach addictive levels, which subsequently fell with a reduction in stress. Their worsening psychological distress may have contributed to the continuation of potential alcoholism in the group that maintained it.

We found a significant interaction between the AUDIT groups and phases for “offline interaction with familiar people.” Although this item tended to increase between the phases for all the AUDIT groups, it was especially significant in the population whose drinking behavior had worsened or remained problematic. Since the quality of the relationships and interactions with these friends and family cannot be understood from the survey data, it is necessary to examine this in future work.

There was also a significant interaction between the AUDIT groups and phases for “difficulties with work or schoolwork.” Regarding those people with no alcohol problem at phase 1, those who developed potential alcoholism at phase 2 reported more severe difficulties with work or schoolwork at both phases than the other groups. In addition, they showed a significant increase in the K6 scores between phases as opposed to those who maintained no alcohol problem and exhibited decreased K6 scores between the two phases. All the other AUDIT groups at phase 1 showed a decreasing trend from phase 1 to phase 2. However, despite the absence of alcohol-related problems, the psychological distress associated with increased work and academic difficulties during the pandemic may have contributed to developing alcoholism. Therefore, these results indicate the need for early care.

### 4.4. Predictors of Alcohol-Related Problems One Year Later

In addition to examining the factors influencing alcohol-related problems at each phase, we analyzed the predictors of alcohol-related problems one year later. Sociodemographic characteristics such as the male gender, unmarried status, increased household income, higher age, larger social network, and decreased COVID-19 prevention behaviors were identified as predictors of hazardous alcohol use after a year. The only sociodemographic characteristic predicting the presence of potential alcoholism after a year was the male gender; nevertheless, mental health and the living environment and behavioral changes such as severe anxiety, a larger social network, increased exercise, a deterioration of economic status, more difficulties owing to a lack of daily necessities, more unhealthy eating habits, and fewer COVID-19 prevention behaviors were identified as predictors of potential alcoholism after a year. Since a state of emergency was declared at phase 1, high levels of anxiety, the deterioration of the living environment and habits, a lower economic status, insufficient daily necessities, and a poor diet at phase 1 may cause prolonged severe alcohol problems (i.e., habitual drinking as a coping behavior).

A larger social network was a predictor of both hazardous alcohol use and potential alcoholism a year later. However, since loneliness (the UCLA score) was not a significant predictor, this result indicates that the mere size of one’s social network is insufficient to prevent stress if its quality is low. In addition, the relation between social isolation and alcohol-related problems may not be a simple linear association, as the diversity of the social network also contributes to alcohol-related problems [47].

Regarding the finding that fewer COVID-19 prevention behaviors were a predictor of both hazardous alcohol use and potential alcoholism, the reduced opportunities to leave home that would require such prevention behaviors during a state of emergency may have led to increased drinking at home; thus, other factors associated with fewer COVID-19 prevention behaviors must be explored. The result that increased exercise, which improves health, contributed to potential alcoholism one year later was difficult to interpret. The nature of the exercise and related factors (e.g., more drinking with peers after exercise) must therefore be examined. These findings may indicate that people at risk of alcohol-related problems have larger social networks and exercise habits that are challenging to identify.

### 4.5. Limitations of the Study

This research has several limitations. First, we could not assess changes in alcohol use before and after the COVID-19 pandemic because we did not collect data before the outbreak. Second, because the data were collected through an online survey and random sampling could not be conducted, the representativeness of the sample could not be guaranteed; furthermore, the sample could not be matched to the respective percentages by age group and sex in each region. Third, because the participants were registered with the survey company, they may be more motivated to cooperate with the survey than those who were not. This characteristic of the participants may have influenced the results of this study. Fourth, almost 16% of the participants dropped out between the two phases. Moreover, the people who responded at phases 1 and 2 had lower K6, PHQ-9, GAD-7, and LSNS-6 scores and a higher age than those who responded only at phase 1, although there was no significant difference in the AUDIT scores between the groups; these differences could have affected the results of this study. Fifth, as Japan’s COVID-19 policy differed from those in other countries in that there were no enforcements or penalties, a comparison with countries that applied different policies to Japan could provide useful information. Lastly, this study’s ANOVA results should be interpreted with caution because of the small effect size overall.

## 5. Conclusions

During the prolonged COVID-19 pandemic in Japan, mental problems such as psychological distress and anxiety and increased work (or academic) and economic difficulties were associated with severe alcohol problems one year later. However, the finding that greater interaction with friends and family and a larger social network were associated with alcohol use problems indicates the difficulty in discerning the risk of alcohol abuse without a deeper analysis of the quality of those relationships. Nonetheless, this one-year longitudinal study showed the state of alcohol-related problems during the prolonged COVID-19 pandemic in Japan and could thus contribute to the early prevention and follow-up of alcohol-related problems in the future.

## Figures and Tables

**Figure 1 ijerph-20-03871-f001:**
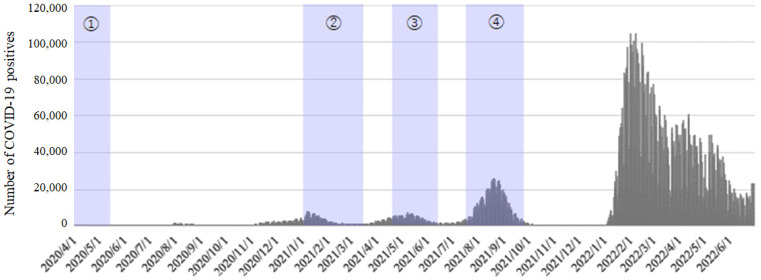
Number of COVID-19-positive cases per day in Japan. ①–④ represent the first to fourth state of emergency declarations in Japan.

**Figure 2 ijerph-20-03871-f002:**
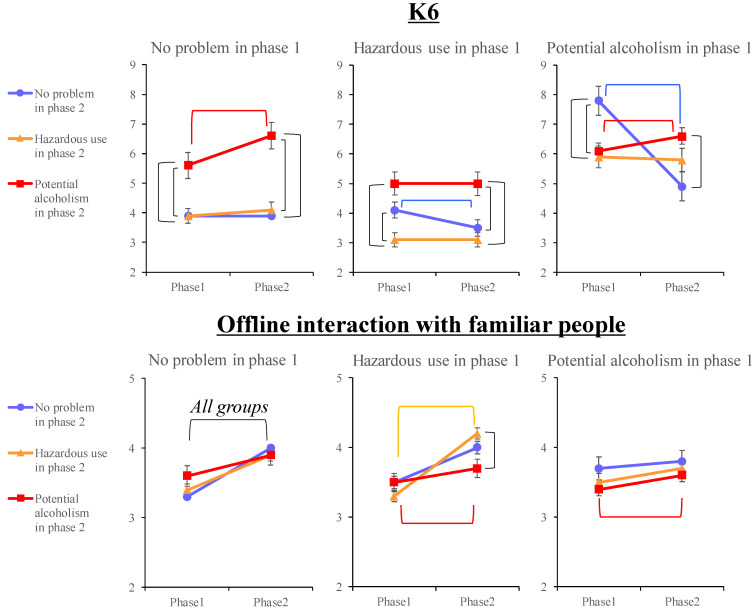
Results of the simple main effect tests of the variables showing significance between the group and phase (*p* < 0.05).

**Table 1 ijerph-20-03871-t001:** Differences in the sociodemographic characteristics between the three AUDIT groups in each phase.

Sociodemographic Indexes at Phase 1	N (%) in Each AUDIT Group at Phase 1						Group Difference			N (%) in Each AUDIT Group at Phase 2						Group Difference		
	No Problem		Hazardous Use		Potential Alcoholism		*χ* ^2^	*p*	Cramer’s *V*	No Problem		Hazardous Use		Potential Alcoholism		*χ* ^2^	*p*	Cramer’s *V*
Overall	7878		1049		687					7785		1131		698				
Sex							327.29	<0.001	0.185							383.99	<0.001	0.200
*Male*	3912	(75.4) −	768	(14.8) +	509	(9.8) +				3826	(73.7) −	846	(16.3) +	517	(10.0) +			
*Female*	3966	(89.6) +	281	(6.4) −	178	(4.0) −				3959	(89.5) +	285	(6.4) −	181	(4.1) −			
Age (years)							87.43	<0.001	0.067							90.35	<0.001	0.069
*20–29*	511	(88.0) +	38	(6.5) −	32	(5.5)				511	(88.0) +	41	(7.1) −	29	(5.0) −			
*30–49*	3437	(84.0) +	383	(9.4) −	270	(6.6)				3404	(83.2) +	391	(9.6) −	295	(7.2)			
*50–64*	2780	(77.9) −	462	(12.9) +	327	(9.2) +				2736	(76.7) −	533	(14.9) +	300	(8.4) +			
*≥65*	1150	(83.7)	166	(12.1)	58	(4.2) −				1134	(82.5)	166	(12.1)	74	(5.4) −			
Occupation							128.17	<0.001	0.082							152.42	<0.001	0.089
*Employed*	5348	(79.4) −	816	(12.1) +	570	(8.5) +				5258	(78.1) −	905	(13.4) +	571	(8.5) +			
*Homemaker*	1307	(91.5) +	82	(5.7) −	40	(2.8) −				1309	(91.6) +	73	(5.1) −	47	(3.3) −			
*Student*	63	(84.0)	7	(9.3)	5	(6.7)				63	(84.0)	9	(12.0)	3	(4.0)			
*Unemployed*	915	(84.6) +	113	(10.4)	54	(5.0) −				913	(84.4) +	112	(10.4)	57	(5.3) −			
*Other*	245	(83.3)	31	(10.5)	18	(6.1)				242	(82.3)	32	(10.9)	20	(6.8)			
Marital status							15.61	<0.001	0.040							9.30	0.010	0.031
*Married*	4956	(80.9) −	725	(11.8) +	443	(7.2)				4903	(80.1) −	759	(12.4) +	462	(7.5)			
*Unmarried*	2922	(83.7) +	324	(9.3) −	244	(7.0)				2882	(82.6) +	372	(10.7) −	236	(6.8)			
Presence of children							17.18	<0.001	0.042							13.24	0.001	0.037
*Yes*	4384	(80.7) −	654	(12.0) +	395	(7.3)				4330	(79.7) −	683	(12.6) +	420	(7.7) +			
*No*	3494	(83.6) +	395	(9.4) −	292	(7.0)				3455	(82.6) +	448	(10.7) −	278	(6.6) −			
Annual household income (JPY)							58.64	<0.001	0.062							66.02	<0.001	0.066
*<2.0 million*	511	(84.5) +	46	(7.6) −	48	(7.9)				511	(84.5) +	52	(8.6) −	42	(6.9)			
*2.0–3.9 million*	1483	(84.3) +	175	(9.9) −	102	(5.8) −				1475	(83.8) +	171	(9.7) −	114	(6.5) −			
*4.0–5.9 million*	1541	(80.6)	241	(12.6)	130	(6.8)				1526	(79.8)	246	(12.9)	140	(7.3)			
*6.0–7.9 million*	1119	(80.4)	150	(10.8)	122	(8.8)				1095	(78.7)	176	(12.6)	120	(8.6)			
*≥8.0 million*	1522	(76.1) −	296	(14.8) +	183	(9.1) +				1489	(74.4) −	329	(16.4) +	183	(9.1) +			

Cramer’s *V*: 0.100 ~ small; 0.300 ~ medium; 0.600 ~ large; +: adjusted residuals ≥ 1.96; −: adjusted residuals < −1.96. JPY: Japanese yen.

**Table 2 ijerph-20-03871-t002:** Number (%) of the participants in each AUDIT group at phases 1 and 2.

	Phase 2						
Phase 1	No Problem		Hazardous Use		Potential Alcoholism		Total
No problem	7283	(75.8)	449	(4.7)	146	(1.5)	7878 (81.9)
Hazardous use	383	(4.0)	484	(5.0)	182	(1.9)	1049 (10.9)
Potential alcoholism	119	(1.2)	198	(2.1)	370	(3.8)	687 (7.1)
Total	7785 (81.0)		1131 (11.8)		698 (7.3)		9614 (100.0)

**Table 3 ijerph-20-03871-t003:** Comparison of each variable between phases 1 and 2.

	Mean (SD)				Difference between the Phases			
	Phase 1		Phase 2		Difference (95% CI)		*p*	Cohen’s *d*
AUDIT	4.20	(5.89)	4.26	(5.88)	−0.06	(−0.14, 0.03)	0.216	0.009
K6	4.11	(5.35)	4.09	(5.44)	0.02	(−0.07, 0.12)	0.604	0.005
PHQ-9	4.10	(5.61)	4.00	(5.58)	0.10	(0.01, 0.19)	0.030	0.018
GAD-7	3.03	(4.52)	2.92	(4.41)	0.11	(0.04, 0.18)	0.002	0.024
LSNS-6	9.45	(6.13)	9.36	(6.05)	0.09	(0.00, 0.18)	0.055	0.015
UCLA-LS3	24.03	(5.85)	24.09	(5.87)	−0.06	(−0.14, 0.02)	0.123	0.011
COVID-19 pandemic-related variables								
*Exercise*	3.70	(1.90)	3.80	(1.88)	−0.10	(−0.14, −0.07)	<0.001	0.054
*Healthy eating habits*	4.15	(1.61)	4.18	(1.61)	−0.03	(−0.06, 0.00)	0.025	0.021
*Healthy sleep habits*	4.66	(1.73)	4.67	(1.68)	−0.02	(−0.05, 0.02)	0.347	0.010
*Favorite activity*	3.66	(1.68)	3.81	(1.69)	−0.15	(−0.19, −0.12)	<0.001	0.090
*Offline interaction with familiar people*	3.35	(1.78)	3.94	(1.77)	−0.59	(−0.63, −0.55)	<0.001	0.334
*Online interaction with familiar people*	2.65	(1.74)	2.78	(1.78)	−0.13	(−0.17, −0.10)	<0.001	0.076
*Continuous prevention behaviors of COVID-19*	5.49	(1.72)	5.48	(1.68)	0.00	(−0.03, 0.04)	0.853	0.002
*Optimism*	4.15	(1.55)	4.22	(1.54)	−0.07	(−0.10, −0.04)	<0.001	0.046
*Deterioration of the household economy*	3.47	(1.71)	3.57	(1.68)	−0.10	(−0.14, −0.07)	<0.001	0.061
*Deterioration of the relationship with familiar people*	2.63	(1.57)	2.69	(1.54)	−0.05	(−0.09, −0.02)	0.003	0.034
*Frustration*	3.18	(1.72)	3.15	(1.66)	0.03	(0.00, 0.07)	0.0497	0.020
*COVID-19-related anxiety*	3.45	(1.67)	3.25	(1.61)	0.20	(0.17, 0.24)	<0.001	0.123
*COVID-19-related sleeplessness*	2.46	(1.51)	2.41	(1.46)	0.06	(0.03, 0.09)	<0.001	0.038
*Difficulties owing to the lack of daily necessities*	2.55	(1.56)	2.49	(1.53)	0.06	(0.02, 0.09)	<0.001	0.037
*Difficulties in* *work or schoolwork*	2.81	(1.70)	2.71	(1.65)	0.10	(0.06, 0.14)	<0.001	0.060

SD, standard deviation; CI, confidence interval; COVID-19, coronavirus disease 2019; K6, Kessler Psychological Distress Scale-6; PHQ-9, Patient Health Questionnaire-9; UCLA-LS3, UCLA Loneliness Scale (version 3); LSNS-6, Lubben Social Network Scale (abbreviated version); Cohen’s *d*: 0.200 ~ small; 0.500 ~ medium; 0.800 ~ large.

**Table 4 ijerph-20-03871-t004:** Scores (standard deviations) of the psychological indexes and COVID-19 pandemic-related variables in each AUDIT group at each phase.

		No Problem at Phase 1						Hazardous Use at Phase 1						Potential Alcoholism at Phase 1					
	Phase	No Problem at Phase 2		Hazardous Use at Phase 2		Potential Alcoholism at Phase 2		No Problem at Phase 2		Hazardous Use at Phase 2		Potential Alcoholism at Phase 2		No Problem at Phase 2		Hazardous Use at Phase 2		Potential Alcoholism at Phase 2	
K6	1	3.9	(5.3)	3.9	(5.1)	5.6	(6.0)	4.1	(4.9)	3.1	(4.4)	5.0	(5.9)	7.8	(6.9)	5.9	(6.2)	6.1	(6.0)
	2	3.9	(5.4)	4.1	(5.3)	6.6	(6.6)	3.5	(4.8)	3.1	(4.5)	5.0	(5.9)	4.9	(5.6)	5.8	(6.0)	6.6	(6.2)
PHQ-9	1	3.9	(5.5)	3.4	(4.8)	5.2	(6.3)	4.1	(5.5)	3.1	(4.6)	5.2	(6.2)	8.1	(8.1)	6.2	(6.8)	6.8	(7.0)
	2	3.7	(5.4)	4.2	(5.7)	7.1	(7.6)	3.4	(4.4)	3.1	(4.6)	5.4	(6.6)	5.8	(6.6)	6.1	(7.1)	7.0	(7.1)
GAD-7	1	2.9	(4.4)	2.5	(3.9)	4.3	(5.3)	2.9	(4.1)	2.4	(3.8)	3.7	(4.8)	6.1	(6.6)	4.6	(5.3)	5.2	(5.8)
	2	2.7	(4.3)	2.7	(4.1)	5.4	(5.7)	2.5	(3.6)	2.3	(3.9)	4.0	(5.3)	4.4	(5.5)	4.3	(5.0)	5.3	(5.6)
UCLA-LS3	1	24.0	(6.0)	23.5	(5.2)	25.0	(4.6)	23.8	(5.4)	22.6	(5.4)	24.4	(5.4)	25.4	(4.0)	25.0	(4.9)	25.5	(5.6)
	2	24.1	(6.0)	23.6	(5.4)	24.9	(3.8)	23.6	(5.4)	22.7	(5.4)	24.8	(5.0)	25.2	(4.6)	25.4	(5.3)	25.6	(5.5)
LSNS-6	1	9.3	(6.1)	10.0	(6.2)	9.4	(6.3)	10.1	(5.8)	10.8	(6.5)	10.3	(6.6)	8.9	(6.7)	9.4	(6.2)	9.6	(6.4)
	2	9.2	(6.0)	10.1	(6.1)	10.1	(6.4)	9.8	(5.7)	10.6	(6.1)	9.5	(6.6)	9.2	(6.0)	9.3	(6.4)	9.1	(6.5)
COVID-19 pandemic-related variables																			
*Exercise*	1	3.6	(1.9)	3.8	(1.9)	3.7	(1.9)	3.9	(1.9)	4.0	(1.9)	3.9	(1.9)	3.9	(1.8)	3.7	(1.8)	3.8	(1.9)
	2	3.8	(1.9)	3.8	(1.8)	4.1	(1.6)	3.9	(1.9)	4.1	(1.8)	3.9	(1.9)	3.9	(1.8)	3.7	(1.9)	3.7	(1.9)
*Healthy eating habits*	1	4.2	(1.6)	4.1	(1.6)	3.9	(1.7)	4.2	(1.5)	4.3	(1.6)	4.0	(1.6)	4.1	(1.6)	4.0	(1.6)	3.9	(1.5)
	2	4.2	(1.6)	4.0	(1.6)	4.0	(1.5)	4.2	(1.6)	4.3	(1.6)	3.9	(1.7)	4.0	(1.5)	3.8	(1.6)	3.8	(1.7)
*Healthy sleep habits*	1	4.7	(1.7)	4.6	(1.7)	4.1	(1.7)	4.7	(1.6)	4.7	(1.7)	4.6	(1.7)	4.2	(1.6)	4.2	(1.7)	4.1	(1.7)
	2	4.7	(1.7)	4.5	(1.7)	4.3	(1.6)	4.7	(1.6)	4.8	(1.7)	4.4	(1.7)	4.0	(1.6)	4.4	(1.6)	4.1	(1.8)
*Favorite activity*	1	3.7	(1.7)	3.6	(1.7)	3.6	(1.7)	3.8	(1.6)	3.9	(1.7)	3.7	(1.7)	3.8	(1.6)	3.6	(1.6)	3.6	(1.6)
	2	3.8	(1.7)	3.8	(1.6)	4.0	(1.5)	3.8	(1.6)	4.1	(1.6)	3.8	(1.7)	3.7	(1.6)	3.5	(1.6)	3.6	(1.7)
*Offline interaction with familiar people*	1	3.3	(1.8)	3.4	(1.7)	3.6	(1.7)	3.5	(1.7)	3.3	(1.8)	3.5	(1.7)	3.7	(1.6)	3.5	(1.7)	3.4	(1.7)
	2	4.0	(1.8)	3.9	(1.7)	3.9	(1.5)	4.0	(1.7)	4.2	(1.7)	3.7	(1.8)	3.8	(1.6)	3.7	(1.7)	3.6	(1.8)
*Online interaction with familiar people*	1	2.6	(1.7)	2.7	(1.7)	3.1	(1.6)	2.6	(1.6)	2.7	(1.8)	2.8	(1.8)	3.3	(1.7)	2.9	(1.7)	3.0	(1.8)
	2	2.7	(1.8)	2.9	(1.8)	3.5	(1.6)	2.8	(1.7)	2.8	(1.8)	2.8	(1.8)	3.4	(1.8)	2.9	(1.7)	3.0	(1.8)
*Continuous prevention behaviors of COVID-19*	1	5.6	(1.7)	5.3	(1.8)	4.5	(1.9)	5.4	(1.7)	5.4	(1.7)	5.2	(1.7)	4.8	(1.7)	4.8	(1.8)	5.0	(1.8)
	2	5.6	(1.6)	5.3	(1.7)	4.6	(1.7)	5.4	(1.7)	5.6	(1.6)	4.9	(1.8)	4.6	(1.8)	4.8	(1.8)	4.9	(1.8)
*Optimism*	1	4.2	(1.6)	4.2	(1.5)	3.9	(1.6)	4.3	(1.4)	4.3	(1.5)	4.2	(1.5)	4.0	(1.6)	3.8	(1.6)	3.8	(1.5)
	2	4.2	(1.5)	4.3	(1.5)	4.3	(1.5)	4.2	(1.5)	4.4	(1.4)	4.1	(1.5)	4.0	(1.6)	4.0	(1.6)	3.9	(1.6)
*Deterioration of the household economy*	1	3.4	(1.7)	3.4	(1.6)	3.6	(1.5)	3.5	(1.6)	3.4	(1.7)	3.6	(1.6)	3.8	(1.6)	3.6	(1.7)	4.0	(1.7)
	2	3.5	(1.7)	3.4	(1.7)	4.0	(1.5)	3.6	(1.6)	3.5	(1.6)	3.6	(1.6)	3.6	(1.5)	3.7	(1.8)	4.0	(1.7)
*Deterioration of the relationship with familiar people*	1	2.6	(1.5)	2.7	(1.6)	3.2	(1.6)	2.8	(1.5)	2.5	(1.6)	3.0	(1.7)	3.6	(1.5)	3.1	(1.7)	3.2	(1.7)
	2	2.6	(1.5)	2.7	(1.5)	3.7	(1.5)	2.8	(1.5)	2.6	(1.5)	2.9	(1.6)	3.2	(1.4)	3.1	(1.6)	3.1	(1.6)
*Frustration*	1	3.1	(1.7)	3.1	(1.7)	3.6	(1.6)	3.3	(1.6)	3.0	(1.7)	3.4	(1.7)	4.0	(1.6)	3.5	(1.7)	3.6	(1.7)
	2	3.1	(1.7)	3.1	(1.6)	3.8	(1.5)	3.2	(1.5)	3.0	(1.6)	3.3	(1.7)	3.7	(1.6)	3.4	(1.6)	3.6	(1.7)
*COVID-19-related anxiety*	1	3.4	(1.7)	3.4	(1.6)	3.6	(1.7)	3.5	(1.6)	3.3	(1.7)	3.4	(1.6)	4.0	(1.6)	3.6	(1.7)	3.6	(1.7)
	2	3.2	(1.6)	3.1	(1.5)	3.8	(1.4)	3.3	(1.5)	3.1	(1.6)	3.3	(1.6)	3.5	(1.4)	3.4	(1.5)	3.4	(1.7)
*COVID-19-related sleeplessness*	1	2.4	(1.5)	2.6	(1.5)	3.3	(1.5)	2.6	(1.5)	2.3	(1.4)	2.7	(1.6)	3.7	(1.7)	3.0	(1.6)	2.9	(1.6)
	2	2.3	(1.4)	2.6	(1.5)	3.6	(1.5)	2.5	(1.4)	2.2	(1.4)	2.7	(1.5)	3.1	(1.5)	2.9	(1.5)	2.8	(1.7)
*Difficulties owing to the lack of daily necessities*	1	2.5	(1.5)	2.7	(1.6)	3.3	(1.5)	2.8	(1.5)	2.4	(1.5)	2.8	(1.5)	3.7	(1.7)	3.0	(1.6)	3.1	(1.7)
	2	2.4	(1.5)	2.7	(1.5)	3.6	(1.6)	2.5	(1.5)	2.3	(1.4)	2.9	(1.6)	3.1	(1.5)	2.8	(1.5)	2.9	(1.7)
*Difficulties in work or schoolwork*	1	2.7	(1.7)	2.9	(1.7)	3.3	(1.6)	3.0	(1.7)	2.7	(1.7)	3.1	(1.7)	3.7	(1.6)	3.3	(1.8)	3.4	(1.8)
	2	2.6	(1.6)	2.7	(1.6)	3.8	(1.5)	2.8	(1.6)	2.6	(1.6)	3.0	(1.7)	3.3	(1.6)	3.2	(1.7)	3.2	(1.8)

COVID-19, coronavirus disease 2019; K6, Kessler Psychological Distress Scale-6; PHQ-9, Patient Health Questionnaire-9; UCLA-LS3, UCLA Loneliness Scale (version 3); LSNS-6, Lubben Social Network Scale (abbreviated version).

**Table 5 ijerph-20-03871-t005:** Differences and interactions between the phases and AUDIT groups for the psychological indexes and COVID-19 pandemic-related variables.

	Interaction			Effect of Time			Effect of Group (Phase 1)			Effect of Group (Phase 2)		
	*F*	*p*	*η_p_* ^2^	*F*	*p*	*η_p_* ^2^	*F*	*p*	*η_p_* ^2^	*F*	*p*	*η_p_* ^2^
K6	6.56	<0.001	0.003	4.79	0.029	0.000	36.70	<0.001	0.008	18.51	<0.001	0.004
PHQ-9	2.31	0.055	0.001	0.04	0.842	0.000	50.04	<0.001	0.010	23.10	<0.001	0.005
GAD-7	1.92	0.105	0.001	3.01	0.083	0.000	44.32	<0.001	0.009	25.69	<0.001	0.005
UCLA-LS3	1.24	0.290	0.001	0.77	0.379	0.000	16.41	<0.001	0.003	9.20	<0.001	0.002
LSNS-6	1.99	0.093	0.001	0.93	0.335	0.000	4.87	0.008	0.001	2.80	0.061	0.001
COVID-19 pandemic-related variables												
*Exercise*	1.75	0.137	0.001	0.93	0.334	0.000	2.33	0.099	0.000	0.03	0.970	0.000
*Healthy eating habits*	1.99	0.092	0.001	1.93	0.165	0.000	3.75	0.023	0.001	4.59	0.01	0.001
*Healthy sleep habits*	1.53	0.190	0.001	0.02	0.901	0.000	17.82	<0.001	0.004	6.03	0.002	0.001
*Favorite activity*	0.74	0.566	0.000	9.28	0.002	0.001	4.24	0.014	0.001	0.49	0.613	0.000
*Offline interaction with familiar people*	4.25	0.002	0.002	84.75	<0.001	0.009	0.59	0.555	0.000	0.62	0.538	0.000
*Online interaction with familiar people*	0.92	0.452	0.000	10.06	0.002	0.001	8.05	<0.001	0.002	3.40	0.034	0.001
*Continuous prevention behaviors of COVID-19*	1.89	0.108	0.001	1.85	0.173	0.000	21.57	<0.001	0.004	14.08	<0.001	0.003
*Optimism*	2.32	0.055	0.001	5.35	0.02	0.001	11.55	<0.001	0.002	1.68	0.187	0.000
*Deterioration of the household economy*	2.33	0.054	0.001	3.65	0.056	0.000	3.97	0.019	0.001	7.99	<0.001	0.002
*Deterioration of the relationship with familiar people*	2.17	0.070	0.001	0.21	0.650	0.000	20.85	<0.001	0.004	16.47	<0.001	0.003
*Frustration*	0.95	0.435	0.000	3.41	0.065	0.000	16.32	<0.001	0.003	12.46	<0.001	0.003
*COVID-19-related anxiety*	2.24	0.062	0.001	28.18	<0.001	0.003	7.01	<0.001	0.001	5.31	0.005	0.001
*COVID-19-related sleeplessness*	1.42	0.224	0.001	6.86	0.009	0.001	33.83	<0.001	0.007	21.55	<0.001	0.004
*Difficulties owing to the lack of daily necessities*	0.64	0.634	0.000	7.96	0.005	0.001	27.42	<0.001	0.006	23.17	<0.001	0.005
*Difficulties in work or schoolwork*	2.91	0.020	0.001	8.98	0.003	0.001	19.14	<0.001	0.004	16.25	<0.001	0.003

COVID-19, coronavirus disease 2019; K6, Kessler Psychological Distress Scale-6; PHQ-9, Patient Health Questionnaire-9; UCLA-LS3, UCLA Loneliness Scale (version 3); LSNS-6, Lubben Social Network Scale (abbreviated version). Interaction: interaction among time, AUDIT group at phase 1, and AUDIT group at phase 2.

**Table 6 ijerph-20-03871-t006:** Results of the multinomial logistic regression analysis.

	Hazardous Use (Phase 2)					Potential Alcoholism (Phase 2)				
Predictor (Phase 1)	*β* (SE)		OR [95% CI]		*p*	*β* (SE)		OR [95% CI]		*p*
Sex (ref: female)										
*Male*	0.65	(0.10)	1.92	[1.59–2.32]	<0.001	0.66	(0.13)	1.93	[1.49–2.50]	<0.001
Marital status (ref: married)										
*Unmarried*	0.23	(0.10)	1.26	[1.04–1.53]	0.018	0.07	(0.13)	1.08	[0.83–1.39]	0.578
Annual household income (JPY) (ref: ≥8.0 million)										
*6.0–7.9 million*	−0.19	(0.12)	0.82	[0.65–1.04]	0.107	−0.08	(0.16)	0.93	[0.68–1.27]	0.636
*4.0–5.9 million*	−0.22	(0.11)	0.80	[0.64–0.99]	0.043	−0.18	(0.15)	0.83	[0.62–1.12]	0.224
*2.0–3.9 million*	−0.48	(0.13)	0.62	[0.48–0.79]	<0.001	−0.25	(0.17)	0.78	[0.56–1.08]	0.134
*<2.0 million*	−0.66	(0.19)	0.52	[0.36–0.76]	<0.001	−0.48	(0.24)	0.62	[0.38–1.00]	0.052
AUDIT group at phase 1 (ref: no alcohol problem)										
*Hazardous use*	2.82	(0.93)	16.69	[13.92–20.02]	<0.001	2.98	(0.14)	19.72	[15.14–25.68]	<0.001
*Potential alcoholism*	4.70	(0.14)	23.98	[18.17–31.64]	<0.001	4.70	(0.15)	110.24	[82.04–148.14]	<0.001
Age	0.01	(0.00)	1.01	[1.00–1.02]	0.016	0.00	(0.01)	1.00	[0.99–1.01]	0.520
GAD-7	0.00	(0.01)	1.00	[0.98–1.02]	0.904	0.03	(0.01)	1.03	[1.00–1.05]	0.034
UCLA	−0.02	(0.01)	0.98	[0.97–1.00]	0.103	0.02	(0.01)	1.02	[0.99–1.04]	0.170
LSNS-6	0.02	(0.01)	1.02	[1.01–1.04]	0.005	0.03	(0.01)	1.03	[1.01–1.05]	0.009
Exercise	0.00	(0.03)	1.00	[0.95–1.05]	0.954	0.09	(0.04)	1.09	[1.01–1.17]	0.021
Healthy eating habits	−0.01	(0.03)	0.99	[0.93–1.06]	0.860	−0.15	(0.05)	0.86	[0.79–0.94]	<0.001
Continuous prevention	−0.09	(0.03)	0.91	[0.87–0.96]	<0.001	−0.11	(0.03)	0.90	[0.84–0.96]	0.001
Deterioration of economic status	0.01	(0.03)	1.01	[0.96–1.07]	0.709	0.08	(0.04)	1.09	[1.01–1.17]	0.029
COVID-19-related anxiety	0.04	(0.03)	1.04	[0.98–1.11]	0.236	−0.08	(0.04)	0.92	[0.85–1.01]	0.069
Difficulties owing to the lack of daily necessities	0.00	(0.03)	1.00	[0.94–1.07]	0.883	0.13	(0.04)	1.13	[1.04–1.23]	0.004

*R*^2^ = 0.32 (Cox–Snell), and 0.44 (Nagelkerke). Model *χ*^2^(36) = 2978.63, *p* < 0.001. COVID-19, Coronavirus disease 2019; GAD-7, Generalized Anxiety Disorder-7; UCLA-LS3, UCLA Loneliness Scale Version 3; LSNS-6, abbreviated Lubben Social Network Scale; JPY, Japanese yen.

## Data Availability

The data presented in this study are available on request from the corresponding author.

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
