# Peer review of "Change in Alcohol Use during the Prolonged COVID-19 Pandemic and Its Psychosocial Factors: A One-Year Longitudinal Study in Japan"

_ijerph, 2023, doi:10.3390/ijerph20053871_

Round 1

Reviewer 1 Report

Congratulations!

Author Response

Thank you very much for your reviewing our paper! We will work to improve our paper.

Reviewer 2 Report

I found this paper to be significant, informative, and thorough. It address a true problem for public health: the rate of alcohol consumption during a pandemic and its correlations with psychological distress. I your study accomplished what it set out to do. One question that occurred to me, though, was the nature of increased psychological distress. Are there any studies that indicate that some forms of psychological distress abated over the course of the pandemic? In other words, I think it could be interesting to investigate psychological distress pre- and post-vaccine availability. Anecdotally, I know that some forms of distress have gone from anxiety and fret to tedium. Just a thought. Lastly, I believe there are a couple of places in the document where a word seems to be missing.

Author Response

Dear Reviewer 2

    Thank you very much for your useful comments. We have revised the manuscript considering your suggestions, and the revised text is indicated in red.

Comment:

I found this paper to be significant, informative, and thorough. It address a true problem for public health: the rate of alcohol consumption during a pandemic and its correlations with psychological distress. I your study accomplished what it set out to do. One question that occurred to me, though, was the nature of increased psychological distress. Are there any studies that indicate that some forms of psychological distress abated over the course of the pandemic? In other words, I think it could be interesting to investigate psychological distress pre- and post-vaccine availability. Anecdotally, I know that some forms of distress have gone from anxiety and fret to tedium. Just a thought. Lastly, I believe there are a couple of places in the document where a word seems to be missing.

Response:

    As you commented, it is quite possible that the content of stress has changed today compared to the early phase of the pandemic when the nature of COVID-19 was not known. We also think that this is a very important perspective. When we analyze the longitudinal data including that from the early phase of the pandemic, we should take this into consideration. Thank you so much for your useful comment!

We also think that our paper was not well-worded and was difficult to understand in many places. We have revised the manuscript as much as possible.

Reviewer 3 Report

Abstract and conclusion are not clear & understandable. 

Author should select subjects from outside one country.

Clearly mention the outcome and novelty of your research.

Author Response

Dear Reviewer 3

We appreciate the constructive comments provided. We have revised the manuscript in consideration of your suggestions, and the revised text is indicated in red.

Comment 1:

Abstract and conclusion are not clear & understandable.

Response:

We have revised the abstract and conclusion section as per your comment, by adding more information for readability and clarity.

Comment 2:

Author should select subjects from outside one country.

Response:

The COVID-19 pandemic policy in Japan differed from that of other countries as it was not accompanied by enforcement or penalties. We certainly agree that a comparison with countries that applied different policies than Japan would provide more useful information. However, as we could not include participants from other countries, we added this issue as the fourth limitation to this study (Line 438-441).

Comment 3:

Clearly mention the outcome and novelty of your research.

Response:

    Alcohol use was the main outcome of this study. We have now described it clearly (Line 139). Additionally, based on your comments, we have highlighted the novelty and strength of this study in the Introduction section (Line 93-98).

Reviewer 4 Report

Your article is a complete and very well structured work.

A large study with the participation of a large group of patients.

A correct structure, methodology, very visual graphics that are easily understood.

Being clear from the beginning the question that vertebra all the work.

The relationship between mental problems and alcohol is frequently observed in the literature. An already accepted association between both factors is considered. It was also assumed that an exceptional state such as the pandemic was going to influence the mood with all its consequences. This article proves an assumption that many of us had.

I think that if we add the methodological quality and the interest of the topic, it is an article that should be published.

It would also be potentially interesting to extrapolate this work to other geographical areas and to be able to compare other factors at play.

Your work is an excellent starting point

Author Response

Dear Reviewer 4

Thank you very much for all the positive comments! We will work to improve our paper.

Round 2

Reviewer 3 Report

Manuscript has been revised in good & scientific manner. It is well written.

Author Response

Thank you very much for your useful comments!